# Improving the Robustness of Summarization Models by Detecting and Removing Input Noise

**Kundan Krishna**[1*]    **Yao Zhao**[2]    **Jie Ren**[2]    **Balaji Lakshminarayanan**[2]
**Jiaming Luo**[2]    **Mohammad Saleh**[2]    **Peter J Liu**[2*]
[1]Carnegie Mellon University, work done while at Google Research
[2]Google Research
*Correspondence to: kundank@andrew.cmu.edu, peterjliu@google.com

## Abstract

The evaluation of abstractive summarization models typically uses test data that is identically distributed as training data. In real-world practice, documents to be summarized may contain input noise caused by text extraction artifacts or data pipeline bugs. The robustness of model performance under distribution shift caused by such noise is relatively understudied. We present a large empirical study quantifying the sometimes severe loss in performance – up to 12 ROUGE-1 points – from different types of input noise for a range of datasets and model sizes. We then propose a light-weight method for detecting and removing such noise in the input during model inference without requiring any extra training, auxiliary models, or even prior knowledge of the type of noise. Our proposed approach effectively mitigates the loss in performance, recovering a large fraction of the performance drop, sometimes as large as 11 ROUGE-1 points.

## 1 Introduction

Despite rapid progress in abstractive summarization in recent years (Lewis et al., 2020; Raffel et al., 2020b; Zhang et al., 2020), virtually all works have tested models using test data which is identically distributed as the training data, and little attention has gone into studying their robustness to input distribution shift caused by input noise. Data from different domains which have been addressed in summarization research, may contain noise of different types. For example, when summarizing a news article on a web page, there can be embedded elements such as ads or tweets which may be included as part of the article due to erroneous text extraction. A system summarizing chatroom conversations might encounter artifacts such as URLs, or sometimes even code shared between participants. If the text to be summarized is acquired by scanning a document, noise can be introduced in the form of OCR errors (Jing et al., 2003). However, the impact of different kinds of noise on modern abstractive summarization systems, and ways to accurately detect and remove that noise, remain largely unknown.

In this work, we study how noise in the input affects the output generated by summarization models, and propose a method to detect and remove it. We synthetically inject 4 types of noise to 4 abstractive summarization datasets with diverse styles (Narayan et al., 2018; Kim et al., 2019; Gliwa et al., 2019; See et al., 2017), and quantify the drop in aggregate metrics for the output summaries (Section 3). We also study how the quality of generated summaries varies with factors such as the amount of noise and size of the models. For our experiments, we use PEGASUS (Zhang et al., 2020) models — transformer-based pre-trained models which deliver competitive performance across abstractive summarization benchmarks.

We present a method to detect and remove noisy spans in the input, which works without prior knowledge of the noise type or access to its samples, yet can recover a large fraction of the drop in output quality resulting from noise addition (Section 4). Our approach for detecting noisy spans is based on variations of the *Relative Mahalanobis Distance OOD Score* proposed by Ren et al. (2023), which uses the embeddings computed by the summarization model's encoder. Our approach does not require any additional training or use of external models, hence it is relatively efficient. Figure 1 shows our method's impact on a sample noisy document.

Finally, we investigate how different parts of the model architecture cause the drop in output quality upon adding noise to the input (Section 5). We attribute the performance drop to two phenomena: (i) *corruption* of the representations of non-noisy input tokens computed by the encoder due to contextualization with neighboring noise; and (ii) *dis-*

**Document**: The Office for National Statistics said industrial output fell 0.7% compared with January, when it dropped 0.3%. http://southafrica.co.za/robben-island.html. Unexpectedly warm weather drove the change, because it led to a fall in electricity and gas demand, the ONS said. https://scholarworks.uvm.edu/. Construction output fell by 1.7% in February, down from a revised January reading of zero growth. The construction figure, the biggest drop in nearly a year, was mainly the result of a 2.6% fall in the housebuilding sector. http://www.traveldesigncruises.com/terms-conditions/. Meanwhile, the UK's deficit in goods and services widened to £3.7bn in February, from a revised figure of £3bn in January. https://work.chron.com/there-marketing-jobs-government-22921.html. https://www.italianfilmfestival.com.au/. According to the ONS, the deficit was fuelled by what it called "erratic items", such as imports of gold and aircraft. "The overall trade deficit worsened, but excluding erratic items, the picture improved, as imports fell more than exports," said ONS senior statistician Kate Davies. Howard Archer, chief UK and European economist at IHS Markit, called the figures "a disappointing package of data for the UK economy which fuels suspicion that GDP growth slowed markedly, largely due to consumers becoming more cautious". https://sanejoker.info/en/2017/06/asch-experiment.html. He added: "We suspect UK GDP growth in the first quarter of 2017 slowed to 0.4% quarter-on-quarter from 0.7% quarter-on-quarter in the fourth quarter of 2016 - this would be the weakest growth rate since the first quarter of 2016.

**Summary before noise addition**: "Activity in the UK's industrial and construction sectors slowed in February, according to official figures."
**Summary after noise addition**: "Here are some of the highlights from the latest economic data for the UK."
**Summary after noise filtering**: "Activity in the UK's industrial and construction sectors shrank in February, according to official figures."
**Ground truth summary**: "Activity in the UK's industrial and construction sectors shrank in February, new figures show."

Figure 1: Effect of noise addition and filtering on the model generated summary for a sample document. Random URLs are injected to the original document as noise. The color indicates the value of our proposed OOD score for a text span — red represents positive and blue represents negative OOD scores, with saturation proportional to the magnitude. Removing the detected noisy parts from input and feeding to summarization model results in a summary closer to the ground truth.

*traction* of the decoder such that it assigns non-zero attention to the representations of noisy input tokens. We perform an ablation where we remove the encoder embeddings of the noisy tokens before running the decoder, hence eliminating the effect of decoder distraction. We find that in a majority of cases this leads to partial recovery in output quality suggesting that generally both factors are responsible to some extent for the poor output summaries.

We make the following contributions:

- We quantify the impact of various kinds of noise on pretrained Transformer-based summarization models, demonstrating drops in output quality up to 12 ROUGE-1 points.

- We show that this noise can be detected using adaptations of an out-of-distribution detection technique, without ever being exposed to it in advance. Our approach can recover much of the performance drop (sometimes as large as 11 ROUGE-1 points), improving robustness and safety for real-world model deployment.

- We examine how different parts of the model's computation are affected by the introduction of input noise, leading to generation of inferior summaries.

## 2 Related Work

Research on the behavior of summarization models on noisy inputs is quite sparse. Jing et al. (2003) investigated how extractive summarization models are impacted by OCR errors in scanned documents. More recently, Meechan-Maddon (2019) studied

the effect of noise from ASR errors on CNN based summarization models. In contrast, we experiment with pre-trained Transformer models which are now preferred in popular use due to their superior performance (Lewis et al., 2020; Zhang et al., 2020; Raffel et al., 2020b), and address a wide variety of noise types and summarization datasets. Contemporary to our work, Chen et al. (2023) have studied the impact of misspellings in input to summarization models, whereas our work instead focuses on additive input noise and its explicit removal.

The effect of noisy inputs has also been studied for NLP tasks other than summarization, such as machine translation (Niu et al., 2020) and question answering (Peskov et al., 2019). Multiple works across machine translation (Karpukhin et al., 2019; Vaibhav et al., 2019), question answering (Peskov et al., 2019) and summarization (Jing et al., 2003) have used synthetic noise to create noisy inputs. Similar to these works, we also create synthetic noisy inputs due to lack of a dataset with naturally occurring labeled noise. One distinguishing aspect of our work is that our noise detection/removal method works without exposing the model to the noise during training, which is closer to practical scenarios where unknown types of noise can be encountered after a model is deployed.

## 3 Impact of noise addition

We inject noisy text spans in between sentences of the clean articles. The insert position of each noisy text span is sampled independently and uniformly at random (see Figure 7 in Appendix for an exam-

ple). Overall, we consider the following choices of a noisy text span:

- **Code** - a random line of code from a corpus of Python programs (Husain et al., 2019). Code may be shared in professional chatrooms.

- **Emoji** - randomly sampled emojis taken from the version 15 release on `unicode.org`. Emojis can be found in conversations and social media posts.

- **URL** - a random URL from the first 1% of validation set of the the Colossal Common Crawl Corpus(`C4`) (Raffel et al., 2020b). URLs can be referenced in news articles or mentioned in chatrooms.

- **Randomsent** - a random sentence from the first 1% of validation set of the `C4` corpus.

We experiment with different amounts of noise added to the input which is treated as a hyperparameter. We measure the amount of noise in terms of the number of noisy tokens added to the input divided by the total number of tokens in the input after noise addition. We experiment with 4 different datasets — XSUM (Narayan et al., 2018), CNN/DailyMail (See et al., 2017), SAMSum (Gliwa et al., 2019) and RedditTIFU-long (Kim et al., 2018). Our datasets span a variety of domains, where the first two datasets deal with summarizing news articles, and the remaining two consider summarizing conversations and social media posts respectively. For all experiments with each summarization dataset, we use PEGASUS models (Zhang et al., 2020) finetuned on that dataset. We evaluate the performance of models using ROUGE scores (Lin, 2004) of the corresponding summaries generated by the them.

**Effect of noise amount:** We compare four different levels of noise, 5%, 10%, 25%, and 50% (50% means the amount of noise tokens is equal to the amount of the clean tokens.). As shown in Figure 2, we see a near monotonic decrease in output quality as more noise is added to the data. In Figure 2a, we group it by datasets while averaging across model sizes and noise types. This reveals that some datasets are more robust to noise than others (e.g. CNN/DailyMail is most robust), and the relative trends in performance drops remain similar across different noise amounts. In Figure 2b, we group the performance drops by noise types while averaging across datasets and model sizes. We see

a clear gap between the drops for Code and Randomsent vs Emoji and URL, with the gap widening as the noise amount is increased.

**Effect of noise type:** In general, we see the models are more robust to URLs and emojis, and less robust to Randomsent and Code noise types as demonstrated by performance drops (averaged across model sizes) shown in Figure 2c. We suspect that some of the this could be due to the presence of URLs and emojis in the training dataset itself, due to which the model may have learned to be robust to those noise types. In addition, from Figure 2c we see that models trained on different datasets have varying sensitivity to different kinds of noises. For example, SAMSum is notoriously susceptible to Randomsent noise, leading to a drop of about 10 Rouge-1 points averaged across model sizes (Table 6 in Appendix), whereas for CNN/DailyMail Code is the most harmful type of noise.

**Effect of model size:** We compare PEGASUS models of 3 different sizes (number of parameters) — Small (50M), Base (200M), and Large (500M). As shown by performance drops (averaged over noise types) in Figure 2d, one might expect larger models to be less susceptible to noise, but it does not seem to be the case in general and simply scaling up models may not solve robustness. In some cases, large models can still suffer loss of over 10 ROUGE-1 points with addition of noise (see Table 6 in Appendix).

A qualitative analysis of the summaries generated for noisy inputs revealed that there exist some *frequent* bad summaries which are generated by the models for many noisy inputs. This is observed for models fine-tuned on XSUM and RedditTIFU-long datasets, while for the other two datasets we did not observe such a pattern. We show five of the most frequently generated summaries for XSUM and RedditTIFU-long in Table 1. We see that the generated summary (for noisy inputs) is often just punctuation marks such as a period or a semicolon. Notably, for XSUM dataset, some of the frequently generated bad summaries were also present as ground truth summaries in the train set. For example, *"All images are copyrighted."* was the ground truth summary for 39 articles in the train set. This suggests that upon encountering input noise, the model can fall back to behaving like an unconditioned language model and generating high frequency sequences from the train set.

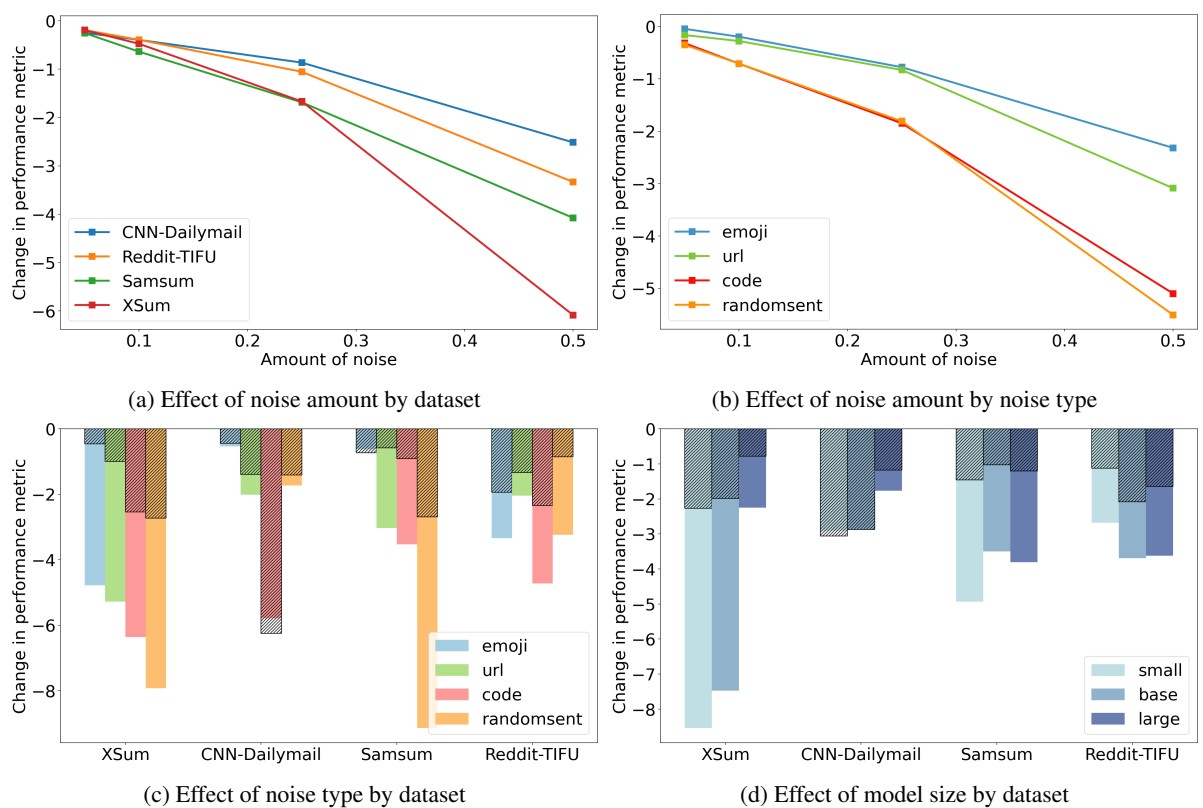

(a) Effect of noise amount by dataset      (b) Effect of noise amount by noise type

(c) Effect of noise type by dataset      (d) Effect of model size by dataset

Figure 2: Change in output quality upon addition of noise to inputs, while varying different factors — noise amount in (a) and (b), noise type in (c), and model size in (d). In (c) and (d) we also show the quality after noise removal (the shaded area). Quality is measured as the geometric mean of ROUGE-1/2/L scores and averaged over the non-varying factors. We set noise amount to 0.5 in (c) and (d).

## 4 Noise detection and quality recovery

### 4.1 Noise detection

Ren et al. (2023) studied various methods for detecting OOD inputs for conditional language generation tasks. They showed that the proposed embedding-based OOD detection method *Relative Mahalanobis distance* (RMD) worked well. Specifically, given an input sequence $\boldsymbol{x} = x_1 \ldots x_t$, the method obtains the input embedding $\boldsymbol{z} = \frac{1}{t}\Sigma_i \boldsymbol{h}_i$ by averaging the encoder's final-layer hidden state vectors $\boldsymbol{h}_i$ corresponding to the input sequence token $x_i$. The OOD score is defined as the difference between two *Mahalanobis distances* (MD),

$$S(\boldsymbol{x}) := \text{RMD}(\boldsymbol{z}) := \text{MD}_{in}(\boldsymbol{z}) - \text{MD}_0(\boldsymbol{z}), \quad (1)$$

where $\text{MD}_{in}(\boldsymbol{z}) = (\boldsymbol{z} - \boldsymbol{\mu})^T \Sigma^{-1}(\boldsymbol{z} - \boldsymbol{\mu})$ measures the distance from $\boldsymbol{z}$ to the fitted in-domain Gaussian distribution $\mathcal{N}(\boldsymbol{\mu}, \Sigma)$, and $\text{MD}_0(\boldsymbol{z}) = (\boldsymbol{z} - \boldsymbol{\mu}_0)^T \Sigma_0^{-1}(\boldsymbol{z} - \boldsymbol{\mu}_0)$ measures the distance to the fitted background Gaussian $\mathcal{N}(\boldsymbol{\mu}_0, \Sigma_0)$. The in-domain Gaussian distribution is fitted using the in-domain training data, and the background distribution is fitted using the same number of examples

from C4 (Raffel et al., 2020a) which represents a broad set of domains. In our experiments we use $10,000$ examples to fit each distribution. The RMD score is regarded as a background contrastive score that indicates how close the input sequence is to the in-domain compared to the background domains. A negative score suggests relatively in-domain, while a positive score suggests OOD.

Instead of computing a single OOD score for the entire input document sequence as in (Ren et al., 2023), in this work, we focus on detecting smaller sub-parts of OOD noise within the input document sequence. We propose three variants:

**Leaveout-Sentence (LO-Sent)** In this case, we compute the OOD scores of the input with and without a sentence in it. The negative of the change in the OOD score after removing the sentence denotes the OOD score of that sentence. Intuitively, if removing the sentence decreases the overall OOD score, that sentence is assigned a positive OOD score and vice-versa.

$$S_{\text{LO-Sent}}(\boldsymbol{x}_{i:j}) = S(\boldsymbol{x}_{1:t}) - S(\boldsymbol{x}_{1:(i-1);(j+1):t})$$
$$(2)$$

| | XSUM | | | RedditTIFU-long | | |
|---|---|---|---|---|---|---|
| Summary | Noisy | Clean | Summary | | Noisy | Clean |
| . *(period)* | 145 | 1 | : *(colon)* | | 230 | 0 |
| A chronology of key events: | 108 | 0 | ** | | 68 | 2 |
| All images are copyrighted. | 62 | 7 | i'm a f**king idiot. | | 16 | 3 |
| All pictures are copyrighted. | 9 | 4 | i'm an idiot. | | 15 | 22 |
| The following is a summary of key events: | 5 | 0 | ] | | 13 | 0 |

Table 1: The frequencies of most commonly generated summaries on noisy versions of XSUM and RedditTIFU-long validation sets (*Noisy*) and their frequencies before adding noise (*Clean*) (using the base model size and Code noise type with noise amount set to 0.5)

**Leaveout-Token (LO-Tok)** This is very similar to the previous method LO-Sent except that instead of removing a sentence, we remove a token at a time and hence get OOD scores for each token,

$$S_{\text{LO-Tok}}(x_i) = S(\boldsymbol{x}_{1:t}) - S(\boldsymbol{x}_{1:(i-1);(i+1):t}). \quad (3)$$

**Sentencewise (Sent)** Instead of computing the score based on embeddings averaged over the tokens in the whole input document sequence (consisting of multiple sentences), we fit Gaussian distributions at the sentence level by averaging the token embeddings in a sentence $\boldsymbol{z}_{i:j} = \frac{1}{j-i+1} \sum_{k=i}^{j} \boldsymbol{h}_k$. We use the sentence embeddings from in-domain data and C4 data to fit the two Gaussian distributions, $\mathcal{N}(\boldsymbol{\mu}^{\text{sent}}, \boldsymbol{\Sigma}^{\text{sent}})$ and $\mathcal{N}(\boldsymbol{\mu}_0^{\text{sent}}, \boldsymbol{\Sigma}_0^{\text{sent}})$.

$$S_{\text{sent}}(\boldsymbol{x}_{i:j}) = \text{MD}_{in}^{\text{sent}}(\boldsymbol{z}_{i:j}) - \text{MD}_0^{\text{sent}}(\boldsymbol{z}_{i:j}) \quad (4)$$

where $\text{MD}_{in}^{\text{sent}}$ and $\text{MD}_0^{\text{sent}}$ are MDs to $\mathcal{N}(\boldsymbol{\mu}^{\text{sent}}, \boldsymbol{\Sigma}^{\text{sent}})$ and $\mathcal{N}(\boldsymbol{\mu}_0^{\text{sent}}, \boldsymbol{\Sigma}_0^{\text{sent}})$ respectively.

**GPT-2 likelihood** We also experiment with a simple language model baseline to generate the noisiness scores based on average negative log-likelihood (NLL) of tokens in a sentence, as given by the pretrained GPT-2 model. Intuitively, a higher value of NLL signifies that a token is unlikely to occur given the past context, which should hold true in case of noisy tokens with clean past context.

$$S_{\text{GPT2}}(\boldsymbol{x}_{i:j}) = -\frac{1}{j-i+1} \sum_{k=i}^{j} \log p_{\text{G}}(x_k|x_{<k}) \quad (5)$$

where $p_{\text{G}}(x_k|x_{<k})$ is the probability assigned by the GPT-2 model to token $x_k$ given previous tokens.

To calculate performance of models at noise detection, we compare the assigned OOD score for each token with its ground truth label and we compute the ROC AUC scores for comparison. For the two sentence level scores, $S_{\text{LO-Sent}}(\boldsymbol{x}_{i:j})$ and $S_{\text{sent}}(\boldsymbol{x}_{i:j})$, we assign each token's OOD score to be the sentence level OOD score for the sentence which contains that token. We compute evaluation metrics in two ways: (i) *per-example* basis where the AUC score is computed for each example and then they are all averaged across the dataset. (ii) *overall* basis where all the predictions across the entire dataset are pooled together before computing a single AUC score. We show the scores averaged across the 4 datasets in (Table 2). In general, the LO-Tok method performs the worst of the three OOD-based methods, while Sent and LO-Sent perform comparably. Comparing the GPT-2 baseline with LO-Tok, GPT-2 performs clearly better for Randomsent, comparably for Code, and clearly worse for Emoji and URL noise types. However, GPT-2 lags behind LO-Sent and Sent for all noise types. Between Sent and LO-Sent, Sent performs better for Code and Randomsent and LO-Sent performs better for Emoji and URL noise types. For its simplicity, we use the Sent method for OOD detection in rest of the paper.

### 4.2 Quality recovery after noise filtering

To remove noise from the input, we simply remove all sentences that have an OOD score greater than a threshold, and then evaluate how much output quality gets recovered after this. We set the threshold of OOD score for filtering to be the 99 percentile value of the OOD scores computed for sentences in the clean version of the dataset (without any noise). The chosen percentile is set to be this high to minimize false positives which can lead to removal of useful non-noisy information from the input. Since the threshold is computed using only the clean dataset and the model trained on that, we do not need any prior information about the noise

| Method | Overall AUC | | | | Per-example AUC | | | |
|--------|------|-------|------------|-------|------|-------|------------|-------|
|        | Code | Emoji | Randomsent | URL   | Code | Emoji | Randomsent | URL   |
| LO-Tok  | 77.10 | 84.25 | 73.63 | 85.41 | 78.52 | 84.17 | 74.74 | 86.83 |
| LO-Sent | 88.04 | 88.83 | 85.43 | 95.66 | 89.46 | 87.94 | 87.00 | 96.08 |
| Sent    | 89.37 | 82.73 | 90.65 | 90.64 | 91.70 | 82.80 | 93.83 | 93.64 |
| GPT-2   | 78.20 | 55.29 | 81.19 | 62.44 | 77.90 | 54.96 | 80.00 | 60.71 |

Table 2: Performance of different methods for noise detection aggregated across datasets (using the base model size and 0.5 noise amount )

(similar to OOD score computation).

We show the performance of noise filtering for different noise types, model sizes and datasets in Table 3. For succinctness, we show the geometric mean of the ROUGE-1,2 and L variants, and point the reader to the Appendix (Table 6) for detailed results with individual variants of ROUGE. After noise filtering, we can recover a large part of the drop in ROUGE scores that occurred due to the added noise. In cases of large drop such as the Randomsent noise type with XSUM and SAMSum datasets, we can recover 4-6 and 6-7 points respectively depending on the model size (Table 3).

We also present aggregate trends of recovery of output quality using our filtering approach in Figure 2c and 2d. We can see that we recover over half of the drop in the performance on 9 out of 16 combinations of datasets and noise types (Figure 2c), with the best performance observed on XSUM and SAMSum datasets and the worst on CNN/DailyMail. The method also succeeds in recovering performance across all 3 model sizes (Figure 2d).

We experimented with various thresholding strategies such as setting thresholds to be constant irrespective of the dataset or model (e.g. 0), or to be equal to a different percentile value (other than 99%) of the OOD scores produced by the model used on clean data. We also tried choosing the optimal threshold based on F1-score of noise detection on a hold-out validation set (assuming a scenario where we have access to labeled noisy samples). We tried 6 thresholding techniques in total, compared in Figure 3a. Setting a constant threshold of 0 provides gains in some cases but in other cases makes the model outputs worse, due to filtering out useful non-noisy content. To prevent this, one can use a very high threshold such a 500 which practically eliminates cases of further drop in performance (Figure 3a), but the performance gains produced in that case are small because less

noise is filtered. The best approach turns out to be setting it be the 99 percentile of the clean data OOD scores, which produces different thresholds for different models, and leads to the highest average gain in output quality among the strategies tried, with minimal cases of further degradation. Surprisingly, optimizing the threshold based on F1-score of noise detection on a validation set also reduces the output quality in many cases, suggesting that F1-score may not be the best predictor for the quality of summary produced after filtering.

We conduct noise filtering for each of our experimental setups (all datasets, noise types and amounts, model sizes) with three thresholds — 0, 200 and 500 and compare the resulting change in summary quality with the precision and recall of the noise detection in Figure 3b. We find that a precision lower than around 0.7 usually leads to a drop in summary quality, even if the recall is nearly perfect suggesting that almost all noise has been removed. This suggests that precision is more important than recall for improving summary quality.

## 5 Investigating causes of loss in performance

There are two distinct mechanisms which can lead to worsening of generated summaries upon addition of input noise. The first is the *corruption* of the encoder's representation of useful clean tokens. The encoder transformer uses self-attention over input tokens to generate their contextualized representations. In cases where noise is present in the input, self-attention can distort the encoder representations of clean tokens. The second mechanism is the *distraction* of the decoder such that it assigns non-zero attention to the noisy tokens' embeddings and this impairs its computation. Even if there is no corruption in the embeddings of clean tokens, the embeddings of noisy tokens can receive non-zero cross-attention from the decoder and influence its generation. If neither of these two phenomenon oc-

| Model size | Clean | Code | | Emoji | | Randomsent | | URL | |
|---|---|---|---|---|---|---|---|---|---|
| | | Add | Filter | Add | Filter | Add | Filter | Add | Filter |
| **XSum** | | | | | | | | | |
| Small | 31.66 | 21.43 | 27.50 | 23.28 | 31.33 | 22.28 | 28.44 | 25.50 | 30.30 |
| Base | 35.18 | 27.64 | 32.01 | 30.03 | 34.49 | 26.28 | 32.32 | 26.87 | 33.97 |
| Large | 37.18 | 35.86 | 36.89 | 36.36 | 36.83 | 31.68 | 35.09 | 35.81 | 36.77 |
| **CNN-Dailymail** | | | | | | | | | |
| Small | 31.96 | 25.27 | 23.37 | 31.24 | 31.46 | 30.01 | 30.38 | 29.69 | 30.39 |
| Base | 33.09 | 26.27 | 25.39 | 32.53 | 32.70 | 31.31 | 31.53 | 30.74 | 31.25 |
| Large | 33.44 | 29.60 | 30.99 | 33.11 | 33.02 | 31.97 | 32.36 | 32.03 | 32.67 |
| **Samsum** | | | | | | | | | |
| Small | 37.96 | 33.00 | 36.80 | 36.83 | 36.73 | 28.11 | 35.18 | 34.17 | 37.31 |
| Base | 39.74 | 36.95 | 38.89 | 39.18 | 38.97 | 31.96 | 37.51 | 36.89 | 39.47 |
| Large | 41.63 | 38.80 | 40.91 | 41.46 | 41.42 | 31.85 | 38.58 | 39.19 | 40.81 |
| **Reddit-TIFU** | | | | | | | | | |
| Small | 15.51 | 11.53 | 13.55 | 12.97 | 15.21 | 13.40 | 14.70 | 13.41 | 14.09 |
| Base | 17.54 | 12.16 | 14.55 | 13.33 | 14.42 | 14.18 | 16.62 | 15.71 | 16.23 |
| Large | 18.15 | 13.33 | 16.06 | 14.89 | 15.76 | 13.92 | 17.32 | 15.96 | 16.88 |

Table 3: ROUGE scores (geometric mean of 1/2/L) on clean input and changes when adding different kinds of noise, and after the noise is filtered out using Sent method (Noise amount: 0.5)

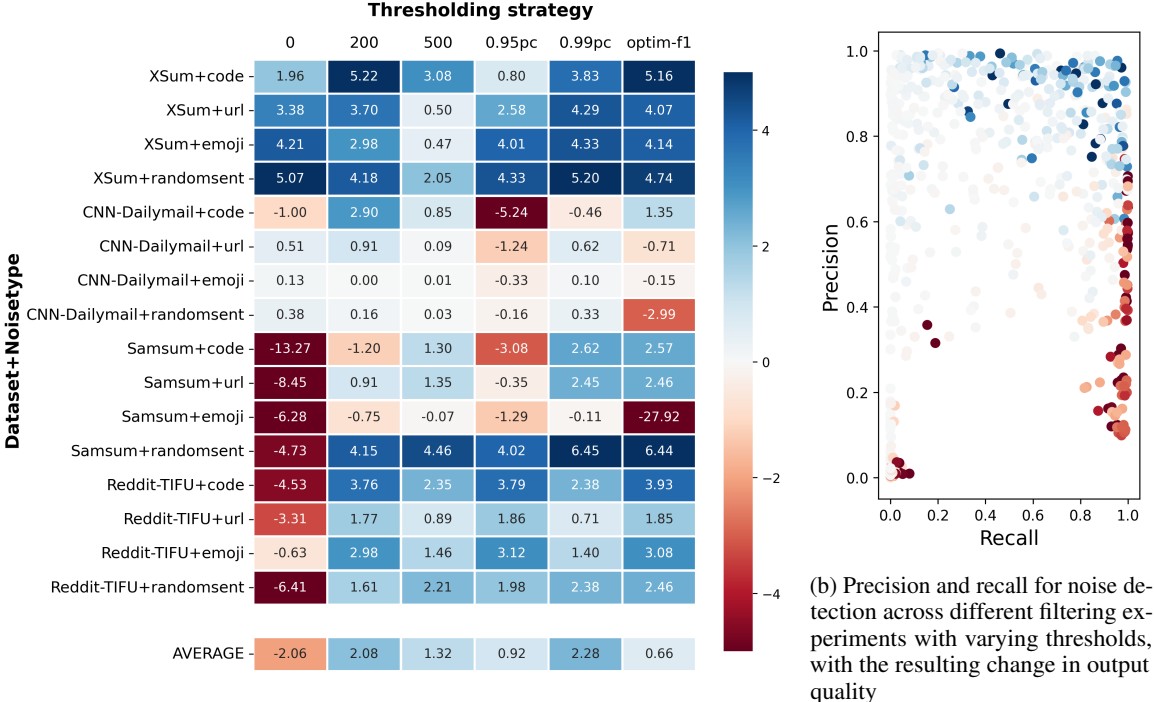

(a) Increase in summary quality after filtering with different thresholding approaches, for different datasets and noise types.

(b) Precision and recall for noise detection across different filtering experiments with varying thresholds, with the resulting change in output quality

Figure 3: Change in output quality for different thresholding techniques (a) and its correlation with the precision and recall of noise detection (b). The changes in summary quality are illustrated by color (blue shows increase and red shows decrease, saturation denotes magnitude clipped to range [0,5])

cur, the generated summary on the noisy and clean variants of any input would be the same. In this section we investigate the contribution of these two factors in the degradation of output quality.

### 5.1 Are the clean token embeddings corrupted by the presence of noise?

We observe that the OOD scores of the clean tokens increase after addition of noise. In Figure 4, we shown an example of this for the XSUM dataset after adding Code noise, where the OOD scores are computed using the Sent method. This suggests that the distribution of clean tokens' embeddings moves farther from the in-domain distribution (learnt from clean in-domain data) relative to the background distribution (learnt from C4 corpus), after adding noise. We observed this for different datasets and noise types, although the extent of the increase in OOD scores varies across them.

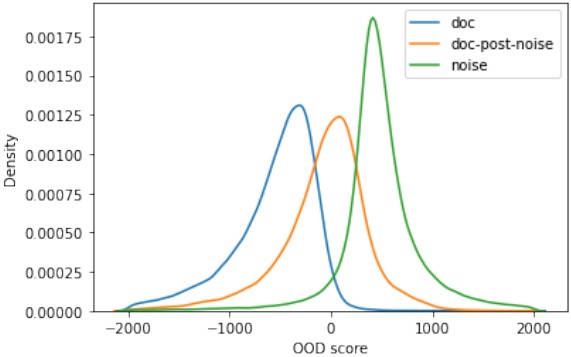

Figure 4: Distribution of OOD scores of (i) clean tokens before adding noise (ii) clean tokens after adding noise and (iii) noisy tokens after adding them (using base model size and 0.5 noise amount)

### 5.2 How much performance can be recovered by preventing distraction of the decoder?

We design an ablation experiment to measure how the performance drop would change if there is no distraction of the decoder by embeddings of noisy tokens. Any drop in output quality in such as setup is attributable only to the corruption of the clean tokens' encoder representations. We remove the embeddings of the (ground truth) noisy tokens after passing the noisy input through the encoder of the PEGASUS model, and then use the decoder to generate the summary using only the remaining embeddings (see Figure 6 in Appendix for detailed workflow). Since the removal is done *after* passing the whole input through the self-attention layers of the encoder, the clean tokens' embeddings are already distorted, and the decoder has to generate the summary using these distorted embeddings. The only difference from the usual scenario is that the decoder does not have to include the noisy tokens' embeddings in the computation. We find that this

mostly leads to an increase in output quality compared to when the noisy token embeddings are not removed (Figure 5). The biggest improvements come for XSUM and SAMSum datasets, whereas for CNN/DailyMail dataset no improvement is seen for any of the 4 noise types. Surprisingly, for the RedditTIFU-long dataset with the URL and Randomsent noise types, removing the noisy tokens' embeddings *decreases* the ROUGE scores further, suggesting that retaining those embeddings is useful for the decoder.

The above ablation study highlights the necessity of running the encoder twice — once for computing OOD scores to detect noise, and then again to compute the encoder representations of the input after removing noisy tokens. While one can save computation time by reusing the encoder embeddings of the clean tokens computed during OOD scoring to feed them to the decoder for generation, results from the ablation suggest that this would give sub-optimal performance recovery (Figure 5).

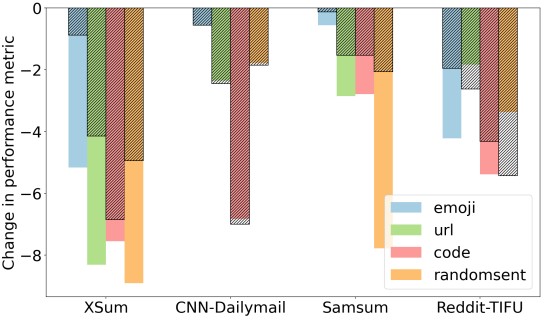

Figure 5: Performance drops for different datasets and noise types, with the shaded area showing drops when the noisy tokens' encoder embeddings are removed before running the decoder (using the base model size and 0.5 noise amount)

## 6 Conclusion and Future Work

In this work, we quantified the impact that noisy inputs can have on the output quality of summarization models, for a variety of datasets and noise types. We then proposed a method to detect and remove noise from the input without using any extra models, training, or prior information about noise types, and demostrated its efficacy. One direction for future work is to investigate what makes certain models more susceptible to specific noise types. Another interesting direction would be to carry out experiments for noise filtering with real-world noisy data rather than using synthetically generated noisy examples.

# 7 Limitations

While we have used 4 different types of noise in our experiments, there can be more types of noise that can be encountered in real world. While a positive aspect of our noise filtering approach is that it can be applied for any unforeseen type of noise, evaluating its performance against all types of noise is infeasible. Due to the heavy compute requirements, we experimented with only one type of pretrained summarization model (PEGASUS), and it is yet to be seen how the results generalize with other models such as T5 (Raffel et al., 2020b) and BART (Lewis et al., 2020). Since our noise detection approach is not perfect, it carries the risk of removing useful information instead of noise. However, our experiments show that while false positives occur, the filtering almost always does more good than harm when applied on noisy documents (Figure 2c). Additionally, the user has an option to minimize the risk of false positives by increasing the threshold of OOD score used for filtering.

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

# A Appendix

## A.1 Selection of shorter inputs to avoid truncation

In our experiments, we exclude those datapoints from the datasets which are longer than a certain threshold. This is done to avoid any truncation of the input (including inputs with added noise) when feeding them into the model. Since adding noise to the input increases its length, it may happen that some clean tokens might be pushed beyond the maximum allowed input length and hence removed when the input is truncated. In such a scenario, removing noisy tokens before feeding the sequence into the model would also cause such clean tokens to be fed into the model again because they can now be accommodated within the input length limit. When measuring the benefit of noise filtering, the benefit from removal of noisy tokens would then be confounded with the benefit from such "resurrection" of clean tokens. To avoid this we only retain those inputs in our datasets where the input length would be within limit even after addition of noise. Since the maximum noise amount we use in our experiments is 0.5, we only retain datapoints which have no more than half of the maximum allowed tokens to input into the model. (Table 4).

| Dataset | Count | Maxlen | Retention |
|---|---|---|---|
| XSUM | 7516 | 512 | 66.5% |
| CNN/DailyMail | 2948 | 512 | 25.6% |
| RedditTIFU-long | 2790 | 512 | 66.2% |
| SAMSum | 686 | 256 | 83.8% |

Table 4: Number of datapoints retained in the test set of datasets after removing inputs longer than maximum length (Maxlen)

## A.2 Compliance with licenses

Among the artifacts used in this work, the PEGA-SUS model and the CNN/DailyMail dataset are distributed under Apache License 2.0, the XSUM and RedditTIFU-long datasets are distributed under the MIT License, and the SAMSum dataset is distributed under CC BY-NC-ND 4.0 License. We use these resources for non-commercial research purposes with proper attribution, which is allowed by all the above licenses.

## A.3 Implementation details

We used PEGASUS models (Zhang et al., 2020) of three different sizes — small, base and large,

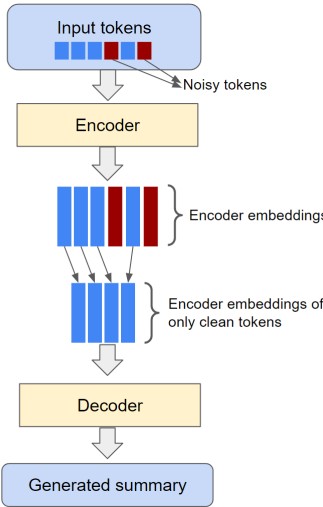

Figure 6: Workflow for the ablation experiment where the decoder does not have to process the noisy tokens' embeddings.

consisting of 50M, 200M and 500M parameters respectively. The experiments. We used TPUs for our experiments. The bulk of the compute expenditure was spent on running inference on different models for generating summaries on various noisy and noise-filtered variants of the datasets. The runtime of each experiment varies with different factors such as model size and dataset size, with the overall estimate for the total compute used at about 700 TPU hours. For model training and summary generation on the XSUM, CNN/DailyMail and RedditTIFU-long datasets, we used the same hyperparameters as used in the original PEGASUS paper (Zhang et al., 2020) and for SAMSum dataset we use the hyperparameters given in Khalman et al. (2021). We used bytefallback during tokenization to enable proper representation of all unicode characters, instead of using UNK tokens. We experimented with a variety of hyperparmeters pertaining to noise addition and filtering, summarized in Table 5. Due to the heavy compute requirements, all our experiments are single run and we did not try multiple seeds for the random noise addition. We used the NLTK[1] library for sentence tokenization and used the rouge_score[2] package from Google Research to compute the ROUGE scores of summaries. The default hyperparameters were used in the ROUGE calculation.

---

[1]https://www.nltk.org/
[2]https://github.com/google-research/google-research/tree/master/rouge

| Hyperparameter | Values |
|---|---|
| Noise type | {Code, Emoji, URL, Randomsent} |
| Noise amount | {0.00, 0.05, 0.10, 0.25, 0.50} |
| Model size | {Small(50M), Base(100M), Large(500M)} |
| Fixed noise filtering threshold | 0, 200, 500 |
| Adaptive noise filtering threshold | 95percentile, 99percentile, optimal-F1(on validation set) |

Table 5: Different hyperparmeters used for noise addition and noise filtering

```
w_end = w_offset + wrg. block2 = superints[idx, :, edgearr[idx, 2]:edgearr[idx, 3]]. McLaren executive director Zak Brown
said he regarded the 17-year-old Englishman as "a fabulous prospect". yield Edge(source=self.resources[identifier_source],.
-----. if getextent:. Norris won two Formula Renault 2.0 titles last season and will move to the European Formula Three
series in 2017. self.shutdown_network(). objectinfo['kmag']),. Triple world champion Lewis Hamilton and Red Bull's Max
```

Figure 7: Sample excerpt from an article from XSum dataset corrupted with code noise.

| Variant | Noise type | ROUGE-1 / 2 / L | | |
|---------|-----------|-----------------|-----|-----|

**XSum**

| Variant | Noise type | Small | Base | Large |
|---------|-----------|-------|------|-------|
| Clean | - | 43.35 / 20.49 / 35.73 | 47.03 / 23.72 / 39.05 | 48.92 / 25.65 / 40.95 |
| Noisy | Code | 31.54 / 12.44 / 25.07 | 38.74 / 17.34 / 31.43 | 47.53 / 24.48 / 39.63 |
| | Emoji | 31.79 / 15.10 / 26.29 | 40.08 / 20.32 / 33.24 | 47.86 / 25.02 / 40.16 |
| | Randomsent | 32.38 / 13.10 / 26.09 | 36.54 / 16.63 / 29.87 | 42.67 / 21.06 / 35.38 |
| | URL | 36.47 / 15.45 / 29.42 | 37.56 / 16.91 / 30.55 | 47.37 / 24.45 / 39.64 |
| Filtered | Code | 38.72 / 17.00 / 31.61 | 43.50 / 20.98 / 35.94 | 48.55 / 25.45 / 40.64 |
| | Emoji | 42.94 / 20.24 / 35.38 | 46.04 / 23.29 / 38.27 | 48.41 / 25.41 / 40.61 |
| | Randomsent | 39.84 / 17.81 / 32.42 | 43.88 / 21.30 / 36.11 | 46.65 / 23.84 / 38.85 |
| | URL | 41.86 / 19.30 / 34.43 | 45.69 / 22.69 / 37.80 | 48.41 / 25.34 / 40.54 |

**CNN-Dailymail**

| Variant | Noise type | Small | Base | Large |
|---------|-----------|-------|------|-------|
| Clean | - | 44.50 / 22.74 / 32.27 | 45.70 / 23.72 / 33.43 | 46.20 / 24.08 / 33.61 |
| Noisy | Code | 36.74 / 16.58 / 26.50 | 38.54 / 17.22 / 27.32 | 42.23 / 20.32 / 30.23 |
| | Emoji | 43.97 / 22.11 / 31.35 | 45.25 / 23.21 / 32.77 | 45.95 / 23.74 / 33.27 |
| | Randomsent | 42.63 / 21.02 / 30.16 | 44.09 / 22.17 / 31.40 | 44.81 / 22.75 / 32.07 |
| | URL | 42.19 / 20.57 / 30.17 | 43.60 / 21.45 / 31.05 | 44.89 / 22.69 / 32.27 |
| Filtered | Code | 33.64 / 15.60 / 24.33 | 36.66 / 16.97 / 26.31 | 43.45 / 21.78 / 31.46 |
| | Emoji | 44.14 / 22.27 / 31.68 | 45.30 / 23.40 / 33.00 | 45.84 / 23.68 / 33.17 |
| | Randomsent | 42.99 / 21.34 / 30.58 | 44.26 / 22.35 / 31.70 | 45.16 / 23.08 / 32.51 |
| | URL | 42.77 / 21.27 / 30.87 | 43.89 / 21.97 / 31.65 | 45.34 / 23.43 / 32.83 |

**Samsum**

| Variant | Noise type | Small | Base | Large |
|---------|-----------|-------|------|-------|
| Clean | - | 50.56 / 25.66 / 42.16 | 51.73 / 27.80 / 43.64 | 53.50 / 29.53 / 45.68 |
| Noisy | Code | 44.81 / 21.32 / 37.62 | 48.32 / 25.29 / 41.30 | 50.24 / 26.85 / 43.29 |
| | Emoji | 49.27 / 24.41 / 41.54 | 50.75 / 27.37 / 43.30 | 53.31 / 29.25 / 45.70 |
| | Randomsent | 39.81 / 17.27 / 32.31 | 42.79 / 21.30 / 35.83 | 42.22 / 21.35 / 35.85 |
| | URL | 46.46 / 22.25 / 38.60 | 48.31 / 25.22 / 41.21 | 50.51 / 27.57 / 43.24 |
| Filtered | Code | 49.22 / 24.56 / 41.24 | 50.70 / 26.94 / 43.07 | 52.43 / 28.87 / 45.23 |
| | Emoji | 49.00 / 24.41 / 41.42 | 50.49 / 27.25 / 43.03 | 53.32 / 29.21 / 45.64 |
| | Randomsent | 47.36 / 23.31 / 39.43 | 49.47 / 25.64 / 41.60 | 50.40 / 26.56 / 42.89 |
| | URL | 49.65 / 25.16 / 41.58 | 51.29 / 27.63 / 43.39 | 52.56 / 28.70 / 45.07 |

**Reddit-TIFU**

| Variant | Noise type | Small | Base | Large |
|---------|-----------|-------|------|-------|
| Clean | - | 24.06 / 7.81 / 19.86 | 26.74 / 9.20 / 21.95 | 27.45 / 9.65 / 22.56 |
| Noisy | Code | 17.95 / 5.74 / 14.87 | 18.72 / 6.21 / 15.46 | 20.35 / 6.91 / 16.85 |
| | Emoji | 20.25 / 6.47 / 16.65 | 20.09 / 7.14 / 16.51 | 22.51 / 7.90 / 18.55 |
| | Randomsent | 21.15 / 6.62 / 17.18 | 22.09 / 7.14 / 18.08 | 21.47 / 7.09 / 17.73 |
| | URL | 21.02 / 6.66 / 17.23 | 24.25 / 8.09 / 19.76 | 24.55 / 8.17 / 20.26 |
| Filtered | Code | 20.98 / 6.83 / 17.37 | 22.24 / 7.58 / 18.27 | 24.31 / 8.46 / 20.15 |
| | Emoji | 23.49 / 7.71 / 19.42 | 21.95 / 7.59 / 17.99 | 23.79 / 8.40 / 19.59 |
| | Randomsent | 23.05 / 7.32 / 18.81 | 25.57 / 8.64 / 20.78 | 26.37 / 9.12 / 21.59 |
| | URL | 21.96 / 7.10 / 17.94 | 24.88 / 8.44 / 20.37 | 25.74 / 8.84 / 21.14 |

Table 6: ROUGE scores on clean input and changes when adding different kinds of noise, and after the noise is filtered out using the Sent method based OOD scores (Noise amount: 0.5)