# OpenReview forum: "Improving the Robustness of Summarization Models by Detecting and Removing Input Noise"
_EMNLP/2023/Conference — EMNLP 2023 Findings_

### Official Review · Reviewer_hpax · 2023-07-20

**Soundness:** 3

**Excitement:**

3: Ambivalent: It has merits (e.g., it reports state-of-the-art results, the idea is nice), but there are key weaknesses (e.g., it describes incremental work), and it can significantly benefit from another round of revision. However, I won't object to accepting it if my co-reviewers champion it.

**Missing References:**

Improving the Robustness of Summarization Systems with Dual Augmentation, ACL 2023

**Paper Topic And Main Contributions:**

The study offers an empirical examination of the significant performance decline due to various input noises across diverse datasets and model sizes. In response, the paper introduces a method that detects and eradicates such noise during the inference phase, eliminating the need for additional training. This method effectively counters the degradation in performance.

**Reasons To Accept:**

1. The presence of noises in datasets is a real concern, and the model introduced effectively addresses the resulting performance dips.

2. The paper investigates the reasons behind performance drop, specifically highlighting the "corruption of the encoder's representation of clean tokens" and the "distraction of the decoder due to non-zero attention assigned to the noisy tokens’ embeddings."

3. Empirical results presented offer valuable insights into how various noise types impact models of different sizes.

**Reasons To Reject:**

1. One of the major concerns lies in the way the noisy experimental dataset has been manually constructed. It can be argued that real-world datasets, especially renowned ones like CNN/DM and XSum, might not exhibit as much noise as has been simulated for this study. This discrepancy could potentially affect the generalizability and applicability of the results.

2. The paper employs a model from Ren et al. (2023) to address the noise issue. However, a more straightforward engineering approach might have sufficed, particularly for common noises like URLs starting with "http" and emojis. These simpler methods could be more efficient and easier to implement in practical applications. Even without identifying the specific categories of noises, these disturbances can still be removed right from the beginning.

3. The paper's results display inconsistency in some scenarios, with performance sometimes witnessing a decline. Notably, there's a drop from 26.27 to 25.39 for the base model after code filtering. The paper would benefit from providing insight into these unexpected outcomes, as currently, readers are left without an understanding of these discrepancies.

**Reproducibility:**

2: Would be hard pressed to reproduce the results. The contribution depends on data that are simply not available outside the author's institution or consortium; not enough details are provided.

**Reviewer Confidence:**

4: Quite sure. I tried to check the important points carefully. It's unlikely, though conceivable, that I missed something that should affect my ratings.

---

> ### Author Rebuttal · Authors · 2023-08-28
>
> Thank you for reviewing our work and providing us with your valuable thoughts. We were delighted to see that you found the empirical results in the submission to be valuable.
>
> We agree that the amount of noise in real-world data may not be as high as what is used in some experiments in the paper. Although we did include experiments with different noise amounts as low as 5% (Figure 2a,b), we used a higher noise amount in our experiments with noise-filtering simply because low noise amounts did not lead to any significant drop in model performance to recover via filtering. However, we believe even demonstrating that low noise amounts lead to minimal damage to model performance is by itself a novel contribution of our work, since to the best of our knowledge there aren’t any studies probing noise-resilience of modern pretrained transformer based summarization models (which are the de-facto standard in today’s NLP world).
>
> Regarding simpler approaches for noise filtering, we weren't able to grasp the proposed approach in the review, and we request you to please elaborate if possible. Based on what we could gather, it seems that maybe you are proposing a pattern-matching based approach to filtering noise, such as filtering out URLs using http prefix or emojis by looking them up from an exhaustive database of emojis. However, there are two problems with this approach:
>
> (1) While pattern-matching can work extremely well for specific noise types, they would need to be handcrafted for different noise types and so it is not a scalable approach. In contrast, our noise filtering approach is generic, works without any prior knowledge of the noise type, and recovers performance across different kinds of noise (Figure 2c).
>
> (2) It is not possible to come up with simple pattern matching approaches for some noise types. For example, the Randomsent noise type in the paper introduces randomly sampled sentences from the C4 corpus as noise, and since it can be any sentence, it is almost surely impossible to come up with a pattern matching approach to distinguish them from the clean sentences. However, since our approach uses contextualized encoder representations for noise detection, we can detect Randomsent noise type with an overall AUC score of 90.65 (Table 2), and after filtering we can recover more than half of the drop in summary quality in 3 out of 4 datasets we experimented with (Figure 2c).

---

### Official Review · Reviewer_PApz · 2023-08-05

**Soundness:** 4

**Excitement:**

4: Strong: This paper deepens the understanding of some phenomenon or lowers the barriers to an existing research direction.

**Paper Topic And Main Contributions:**

The paper studies a practical and novel topic: affect of input noise on abstractive summarization. The study 1) quantifies the impact of various kinds of noise on pretrained Transformer-based summarization models, 2) shows that this noise can be detected using adaptations of recently proposed out-of-distribution detection method 2) examine how different parts of the model’s computation are affected by the introduction of input noise.

**Reasons To Accept:**

This is an interesting and novel task, with practical significance.

A thorough study, showing promising results.

A well written paper that could be of interest to the industry-centered NLP community.


**Reasons To Reject:**

Limited novelty.

The evaluation could be expanded, only PEGASUS models are used on the abstractive summarization task.

**Reproducibility:**

4: Could mostly reproduce the results, but there may be some variation because of sample variance or minor variations in their interpretation of the protocol or method.

**Reviewer Confidence:**

4: Quite sure. I tried to check the important points carefully. It's unlikely, though conceivable, that I missed something that should affect my ratings.

---

> ### Author Rebuttal · Authors · 2023-08-28
>
> Thank you for reviewing our draft and championing our work! We were pleased to see that you appreciated the novelty of the paper’s theme: studying the effect of input noise on modern summarization systems.
>
> For the evaluation, we used only PEGASUS models due to compute resource constraints since we ran a large number of experiments to explore many variations of hyperparameters involved like: (1) model size, (2) noise type (3) noise amount (4) noise filtering strategy. However, exploring the behavior of other pretrained models such as T5 models could be an interesting future direction. Our intuition is that most transformer based models such as T4, BART etc should show similar results since they all have the same architecture and only differ in pretraining strategies, but it is also possible that some pretraining strategies (especially denoising-based pretraining) may lead to higher noise robustness.

---

### Official Review · Reviewer_vmhZ · 2023-08-05

**Soundness:** 2

**Excitement:**

4: Strong: This paper deepens the understanding of some phenomenon or lowers the barriers to an existing research direction.

**Paper Topic And Main Contributions:**

This paper studies the impact of noisy inputs on summarization models and the strategies to remove such noises and provides an analysis of the causes of performance loss from the noisy inputs w.r.t. the encoder and decode of the summarization models.

**Reasons To Accept:**

This paper conducts an in-depth study of the impact of noisy inputs on the summarization models. The authors also present strategies to reduce the impact of such noises and provide an analysis of the cause of performance loss from the input noise. The observations and conclusions of this paper can be useful for understanding the summarization model performance when used in noisy/real-life application scenarios, and the noise-reducing strategies proposed in this paper can also be useful for future work.

**Reasons To Reject:**

The major concern I have is that the input noise is synthetically injected into the datasets in this paper. This mainly leads to two questions/concerns: 1. whether the synthetic noise pattern is similar to the noise pattern from the input of real-world data. 2. whether the noise-reducing strategies proposed in this paper for synthetic noise can also be successfully applied to real-world noisy data. Therefore, personally, I think the observations and conclusions made in the paper are not trustworthy enough when we are considering their effectiveness in real-world application scenarios, and one important missing part of this paper is performing analysis/study on real-world noisy data. For example, since the widely-used CNN/DailyMail and XSum datasets are from news articles on the websites of CNN/BBC, the HTML files that contain the news articles can be a natural source of such noisy input data samples.

**Reproducibility:**

3: Could reproduce the results with some difficulty. The settings of parameters are underspecified or subjectively determined; the training/evaluation data are not widely available.

**Reviewer Confidence:**

4: Quite sure. I tried to check the important points carefully. It's unlikely, though conceivable, that I missed something that should affect my ratings.

---

> ### Author Rebuttal · Authors · 2023-08-28
>
> Thank you for your insightful comments, and for acknowledging the importance of understanding and improving noise-resilience of models.
>
> We agree with your point that it is important to make the method work in real-world noisy data. While the choices of noise types in our study (e.g. URLs) were actually inspired by the plausibility of their occurrence in real world noisy data, we found it infeasible to run large-scale experiments with a real-world noisy dataset due to the following reasons:
> The types of noise that can be encountered by systems deployed in the real world are vast and difficult to capture. Additionally, since the occurrence of noise is rare in natural sources, curating a dataset of real-world noisy inputs would require a significant amount of work which would be a different paper in its own right and outside the scope of our paper. One example is that of “MTNT: A testbed for machine translation of noisy text ”(Michel et al. 2018)  which creates a benchmark with noisy inputs for the machine translation task. On the other hand, for summarization, to the best of our knowledge no such benchmark of noisy inputs exists. This is a recurring challenge in NLP which compels researchers to use synthetic noise due to lack of real-world noisy samples, and multiple published works in the past have used it in machine translation, question answering and summarization (some references provided in lines 127-131).
>
> Hence, to shine light on the unexplored question of how modern transformer-based summarization models would react to noise, we did our best by simulating plausible kinds of noise that are likely to come up in different usage scenarios. For example, we included URLs as they can be frequently shared in chats or articles and could be included in the model input. However, we acknowledge the fact that real-world noise can be diverse and different from the synthetic noise we use, and for that reason we chose to design a noise-filtering approach that works without any prior knowledge about the kind of noise that is present in the input and can recover performance across noise types (Figure 2c).

---

### Meta-Review · Area_Chair_Rijk · 2023-09-17

**Recommendation:** 2

**Metareview:**

The paper studies the impact of noise in input for summarization models, and proposes methods to automatically detect and remove such noise. This is quite a relevant and exciting area for research. The main limitation pointed out by multiple reviewers is the synthetic nature of the noise injected into documents, which raises concerns that the proposed method to detect noise may not generalize to real noise distributions. Already, the noise detection experiments are conducted on a pretty high noise level which the authors agree is not the same as noise levels. I would encourage the authors to collect a small test set of examples with 'real noise' to show generalizability of their approach.

---

### Decision · Program_Chairs · 2023-10-07

**Decision:**

Accept-Findings

**Comment:**

The paper studies the impact of noise in input for summarization models, and proposes methods to automatically detect and remove such noise. This is quite a relevant and exciting area for research. The main limitation pointed out by multiple reviewers is the synthetic nature of the noise injected into documents, which raises concerns that the proposed method to detect noise may not generalize to real noise distributions. Already, the noise detection experiments are conducted on a pretty high noise level which the authors agree is not the same as noise levels. I would encourage the authors to collect a small test set of examples with 'real noise' to show generalizability of their approach.